# Ambiguous Annotations: When is a Pedestrian not a Pedestrian?

Luisa Schwirten
Quality Match GmbH
Heidelberg
ls@quality-match.com

Jannes Scholz
Offenburg University
Offenburg
jscholz@stud.hs-offenburg.de

Daniel Kondermann
Quality Match GmbH
Heidelberg
dk@quality-match.com

Janis Keuper
Offenburg University
Offenburg
janis.keuper@hs-offenburg.de

## Abstract

*Datasets labelled by human annotators are widely used in the training and testing of machine learning models. In recent years, researchers are increasingly paying attention to label quality. However, it is not always possible to objectively determine whether an assigned label is correct or not. The present work investigates this ambiguity in the annotation of autonomous driving datasets as an important dimension of data quality. Our experiments show that excluding highly ambiguous data from the training improves model performance of a state-of-the-art pedestrian detector in terms of LAMR, precision and F1-score, thereby saving training time and annotation costs. Furthermore, we demonstrate that in order to safely remove ambiguous instances and ensure the retained representativeness of the training data an understanding of the properties of the dataset and class under investigation is crucial.*

## 1. Introduction

A crucial, yet difficult task in computer vision for autonomous driving and driver assistance systems is the detection of vulnerable road users, such as pedestrians, cyclists or motorcyclists, and including persons with impaired vision, hearing or mobility. To this day, this group carries the highest risk of injuries and casualties in traffic accidents [7]. Therefore, the development of systems ensuring and improving the protection of these road users is an important step towards enhancing traffic safety for all participants. However, the detection of persons in street scene images is challenging given the individuality and diversity of human appearance.

In the past decade, the capabilities of computer vision systems have seen remarkable progress through the employment of deep learning models, which require vasts amounts of data for training and testing. This data comes in two different forms: (un-annotated) raw data and annotated data. For object detection, the annotations indicate the identity of the objects through class labels, as well as their localization, most commonly in the form of bounding boxes. They can also include further information, such as orientation, or whether the object is partly occluded. Our experiments focus on supervised learning, where the quality of the labelled data is already an important consideration at training time. However, even in the case of unsupervised learning, in order to monitor and ensure the trained model's performance, annotated data as a ground truth is needed. For this reason, annotation quality is crucial for both training regimes, supervised as well as unsupervised. It is possible to synthetically generate ground-truth images for both testing and training, but these lack behind real street scene images in diversity [19][27]. Therefore, images labelled by human annotators are still considered the "gold standard" for ground-truth data.

Human annotation however comes with its own challenges. As humans, we are not immune to errors. A small percentage of the data, even in easy cases, will be labelled incorrectly by human annotators. Moreover, some instances are inherently difficult to label, which often leads to disagreement between different annotators. We refer to images and instances, where the correct label is not entirely obvious as "ambiguous". The following section investigates this ambiguity as an important aspect of data quality for the case of vulnerable road users. The results of our experiments, which are presented in section 3, demonstrate that improved model performance can be achieved by removing highly ambiguous instances from the training set.

## 2. Related Work

Awareness of issues with the reliability of ground truth labels has risen in recent years, marked by publications concerned with the correctness of the annotations in large public datasets and benchmarks, such as ImageNet and CIFAR-10 [3] [14] [16] [20]. At the same time, a large number of publications is concerned with how to handle noisy labels in object classification and detection, and how to train networks, which are robust against noise [2] [1] [11] [15] [21] [23] [24] [26]. However, the definition of label noise used in this field of research implies that there is an underlying true label, which can be observed. Due to the challenges detailed in the following sections and the resulting subjectivity in the labelling of difficult tasks, this is not always the case. In comparison to studies on label noise, a much smaller number of publications exists, which is concerned with incorporating ambiguity information into the training data. Starting with Gao et al. (2017) [10], neural networks have been employed in distribution learning models, to learn a label distribution instead of binary or multi-class labels. Distribution learning approaches do not always utilize the variability in the annotations to derive the ground truth distributions, but oftentimes they are modeled implicitly from neighboring classes [5] [13], from features extracted by a neural network [29], or most recently, using transformers [28]. A reason for this is that information on the variability of annotator answers is not readily available for most datasets [9].

## 3. Ambiguity in Detection Data

### 3.1. Definition

Labelling vulnerable road users in street scene images is not a simple task, and therefore involves a high level of ambiguity. This ambiguity arises from several challenges, making it difficult to recognize instances as their correct class, or to distinguish them from their neighboring classes. Instances, which are partly or even heavily occluded are hard to detect for human annotators as well as machine learning models. The same is true for objects with low visibility, such as blurry instances or those that are far from the camera. The wide variety of lighting conditions found in street scenes poses an additional challenge. This leads to an ambiguity in the images where the true label is not always observable. Even when annotated by experts and in the absence of errors, the assigned labels will therefore retain a degree of subjectivity. This problem has already been described for the field of medical images as "inter-observer variability" [15]. Another term often used in the literature is "label noise" [1], which usually implies that there is a correct label observable from the data, and annotator answers deviating from it are incorrect and add noise to the annotation. In contrast to this, we define ambiguous data as any

instances, where different annotators will disagree on what label to assign, because the true label is not entirely objectively observable. On the example of the class "pedestrian", we further examine the sources of this ambiguity in the following. These can be found in properties of the image itself, or different possible interpretations of the class definitions in the labelling guide, *i.e.* the instructions, which are given to the annotators when labelling the images.

### 3.2. Sources of Ambiguity

**Image Properties**   Ambiguity can arise from the image itself, if the visibility of the instance is impaired due to adverse weather, blurriness or low contrast in the image, partial occlusion by another object, or the object being far away from the camera. This form of ambiguity will always exist in street scene images, which are taken from a vehicle driving outside of controlled conditions "in the wild". Figure 1 shows examples of this for the class "pedestrian". While in the image in 1a the person is easily identifiable, in 1b classification of the instance is much more difficult. Image 1c shows an instance which is highly ambiguous due to low visibility. Without additional data, such as tracking of the person throughout an image sequence, it is in this case impossible to tell with certainty, if in reality this is the image of a person or not. However, additional information, which would help us distinguish between 'real' pedestrians and other object classes is usually not given in publicly available datasets, and not always recorded during the capturing of the images. Images 1d to 1f illustrate how occlusion, which is a common challenge in image annotation, caused different degrees of ambiguity.

**Class Definitions**   In addition to image properties, another common cause of ambiguity is that of instances falling in between the definitions of neighboring classes in the labelling guide, *e.g.* a person could be either labelled a pedestrian or cyclist depending on whether and how they are using a bike. To some extent, this can be managed by covering many possibilities in the labelling instructions. However, even the most detailed class description will not be able to cover all possible cases, especially for such a diverse class as pedestrians. We illustrate this issue using examples from the neighboring classes 'pedestrian' and 'rider' of *e.g.* a bike, motorbike or scooter. Very often, the distinction between the two is made such, that persons who are walking or standing, are to be labelled as pedestrians, while someone riding a bike or scooter is classified as a rider. So a person who is only holding or pushing a bike, but not currently riding one in the image, is by this definition a pedestrian and not a rider. But then what about someone who is sitting on the bike (*i.e.* strictly speaking not walking or standing), but for example waiting at a traffic light, should they be considers as a rider or a pedestrian? Since this is a very common

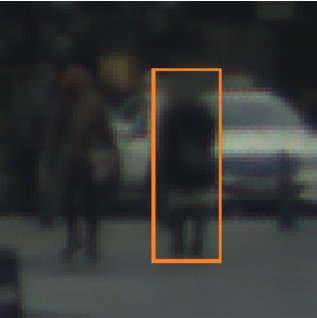

(a) Low ambiguity: A well recognizable pedestrian instance.

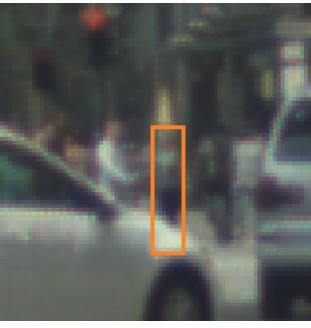

(b) Medium ambiguity: This pedestrian is already harder to identify.

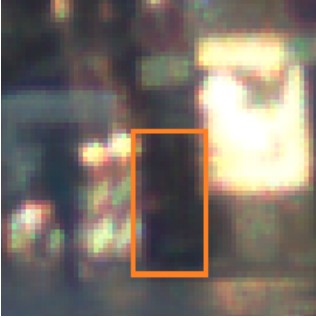

(c) High Ambiguity: It is very hard to identify this instance.

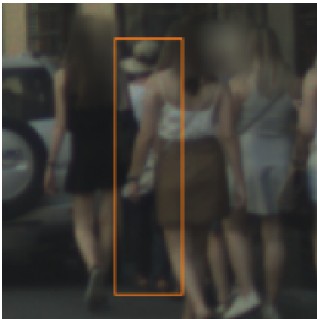

(d) Low ambiguity with occlusion.

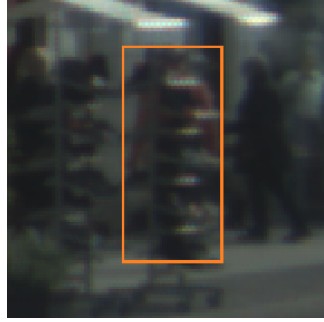

(e) Medium ambiguity with high occlusion.

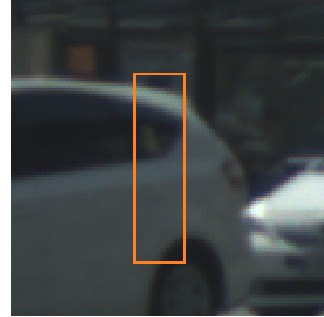

(f) High ambiguity caused by high occlusion.

Figure 1. Image Properties. Medium and high ambiguity corresponds to an ambiguity measure of 0.4 to 0.49 and over 0.65 respectively. Examples from the ECP Dataset [4].

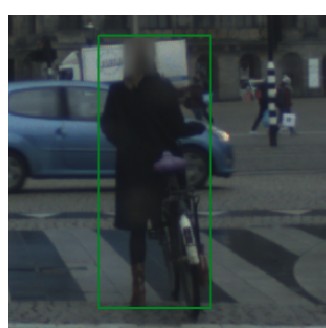

(a) Low ambiguity: A pedestrian with a bike and one foot clearly on the ground.

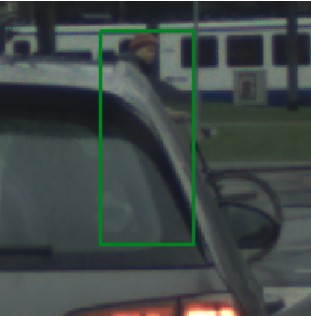

(b) Medium to high ambiguity: This rider could also be interpreted as a pedestrian pushing a bike.

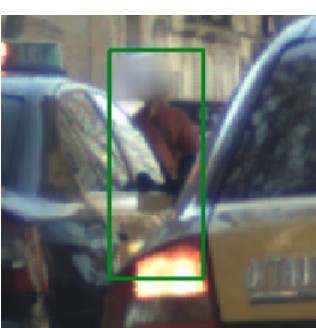

(c) High ambiguity: It is not possible to tell from the image whether the person is on a bike or scooter.

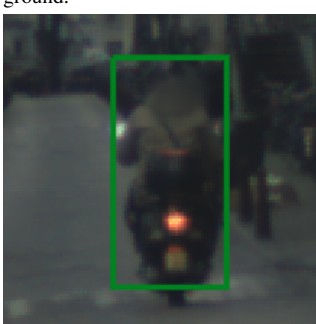

(d) Low ambiguity: A rider with both feet on the vehicle.

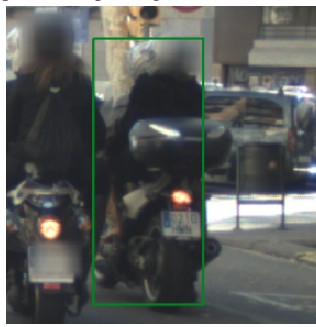

(e) Medium ambiguity: The right foot of the person, which is on the ground, is barley visible.

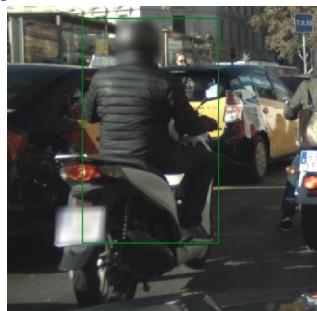

(f) High ambiguity: We can not tell form the image whether or not the left foot is on the ground.

Figure 2. Neighboring Classes: Pedestrian versus Rider, with the distinctive criterion that a person with at least one foot on the ground is to be labelled as a pedestrian. Examples from the ECP Dataset [4].

edge case, the widely adapted distinction here is that anyone who has at least one foot on the ground is to be labelled as "pedestrian". However, this brings us to the next problem, because it is not always clear, whether or not this is the case in an image. Figure 2 illustrates this on six examples. According to the above distinction, the person in the image in 2a is clearly identifiable as a pedestrian, because they have one foot on the ground. Following the same rule, the instance in 2d is to be labelled a rider, because both feet are on the vehicle. The remaining four images exhibit different degrees of ambiguity w.r.t. this distinction. In images 2b and 2c it is not clear, whether or not the person is on a vehicle. In images 2e and 2f it is difficult to tell if the criterion of one foot being on the ground is met or not, *i.e.* if this is a pedestrian or rider per the definition. For these instances, we can expect disagreement between the annotators which of the two neighboring classes to assign.

If we include such cases in the labelling guide as well, *e.g.* by always deciding for one of the two classes, if the legs are not both visible, we will be able to cover more such instances with the instructions, but we will never be able to come up with a finite set of rules that is able to cover all imaginable cases. Moreover, we will want to keep our instructions as concise as possible, because the longer the labelling guide gets, the more this itself can become a source of annotation errors. At some point, the annotators will not be able to correctly remember all the rules we have laid out for them during the process of the annotation. So there will always be some remaining edge cases which fall in between two neighboring classes, and might be labelled differently depending on the annotators' interpretations. Awareness of these challenges and possible pitfalls is crucial, when making decisions w.r.t. the labelling instructions and class definitions.

Since for these reasons a certain degree of ambiguity is inevitable when annotating a dataset, should all these instances be treated identically during training, regardless of their different degrees of ambiguity? And how are highly ambiguous cases to be handled during testing and evaluation of a trained model? Should, for example, the model receive an equally high penalty for not finding the instance in Figure 1c in as it should for not correctly detecting the person in 1a? As a cost-efficient measure, we investigate the effects of simply removing highly ambiguous instances from data.

## 4. How Does Ambiguity Influence the Model Performance?

### 4.1. Model and Training

Our experiments were conducted using data from the EuroCity Persons Dataset (ECP) [4], which is a prominent benchmark for pedestrian detection. Since the test dataset

| Subset | Height | Occlusion | Truncation |
|--------|--------|-----------|------------|
| reasonable | $> 40$ px | $< 40\%$ | $< 40\%$ |
| small | $30 - 60$ px | $< 40\%$ | $< 40\%$ |
| occluded | $> 40$ px | $40 - 80\%$ | $< 80\%$ |
| all | $> 20$ px | $< 80\%$ | $< 80\%$ |

Table 1. ECP Evaluation Subsets

of the benchmark is not publicly available, we used the published validation set as our test set. We chose Pedestron [12] for evaluation, the highest performing model from the benchmark, for which the full architecture as well as pretrained weights are published. This is a Cascade R-CNN model [6], originally with an HRNet [25] backbone, which we replaced with MobileNetV2 [22] to achieve still close-to benchmark performance, but at greatly reduced training times. Each model was trained for 50 epochs, which took approximately 4 days on a single NVIDIA RTX 4090. The reasoning for stopping the training early and choosing a more light-weight backbone was to enable us to train more iterations of the model in the same time, since we were interested in the comparative performance of the model trained on different data, instead of reaching peak performance. We could however confirm that, while training the model for 100 more epochs still lead to minor performance gains, the comparative results between the models stayed the same. The performance of the trained models was evaluated using the official evaluation measure of the ECP benchmark, Log Average Miss Rate (LAMR) [8]. In short, the LAMR expresses the trade-off between the miss rate (ratio of ground truth pedestrians that were not detected) and false positives per image (other objects the model falsely detected as pedestrians) for different thresholds of confidence scores returned by the model.

### 4.2. Measuring Ambiguity

In order to analyse the effects of ambiguous data on training and testing, we need a way to quantify ambiguity within the annotations. For our experiments, we focused on the annotation question whether the instance under consideration is a human being. Annotators are asked to respond to this question with either "yes" or "no", or indicate that they are unable to give a definite answer (denoted "?" in the following). The frequencies of these answers for a given task, $n^{\text{yes}}$, $n^{\text{no}}$ and $n^{?}$, are then used to calculate a heuristic measure for ambiguity from annotator disagreement [17], which defines the ambiguity $\alpha$ of an instance as

$$\alpha = \begin{cases} 1 - \gamma \cdot 2|\frac{n^{\text{yes}}}{n - n^{?}} - \frac{1}{2}| & \text{if } n - n^{?} > 0 \\ 1 & \text{otherwise} \end{cases} \quad (1)$$

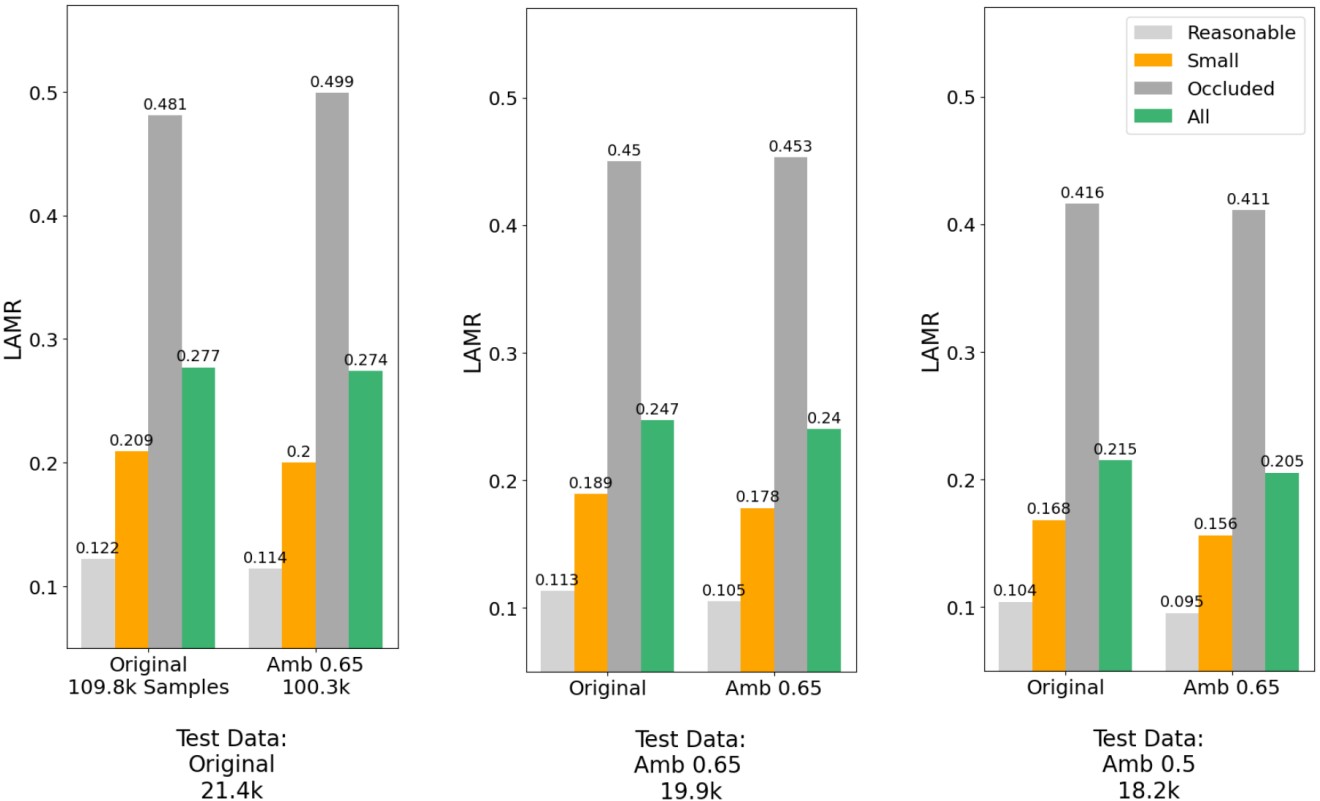

Figure 3. Results for two training sets and three test sets including different degrees of ambiguity. "Original" denotes the original ECP training and validation sets, "Amb 0.65" and "Amb 0.5" the same subsets pruned above an ambiguity threshold of 0.65 and 0.5.

with

$$n = n^{\text{yes}} + n^{\text{no}} + n^{?},$$

where $\gamma \equiv 1 - \frac{n^{?}}{n}$ re-scales the distance of the observed distribution of answers from a uniform distribution by the ratio of "?"-answers, such that, if only "?"-answers are given by the annotators for the task, the ambiguity reaches its maximum value of 1.

For ECP, only hard labels with no information on annotator disagreement exist within the published benchmark data. As a cost-efficient alternative to re-annotating the entire training and validation sets with multiple annotators for the above question, we employed the approach proposed by [18] to estimate the answer distributions. The model pretrained on the ECP Dataset has been proven to estimate annotator answers for ECP with high accuracy [18]. The ambiguity measure was then computed from the predicted answer distributions.

To compare the effects of ambiguity on both, training and test set, we removed highly ambiguous instances up to different ambiguity thresholds from the dataset and trained the model on the entire original data as well as on the versions of the dataset with applied ambiguity thresholds. We then evaluated the trained models on test data including instances, again up to different ambiguity thresholds. The re-

sults for two models, one trained on all original data, and one trained without instances with ambiguity score $\geq 0.65$, which were then tested on three different version of the test set (all data vs. ambiguity thresholds of 0.65 and 0.5), are shown in Figure 3. "Reasonable", "Small", "Occluded", and "All" are the original subsets of the ECP benchmark for evaluation (see Table 1).

### 4.3. Results

**Removing ambiguous data from the training dataset improves model performance.** Figure 3 shows that the model trained without highly ambiguous instances achieves higher performance (lower is better for the LAMR), except when heavily occluded instances are included in the evaluation. Upon further investigation of the prediction results, we found that the reason for this better performance is, that for instances up to moderate occlusion, removing ambiguous instances from the training set improves precision at the expense of only a small decline in recall. Precision, Recall and F1-Score for the two different training regimes when tested on data with and without high ambiguity are given in Figure 4. We can see that the model trained without highly ambiguous data also performs better in terms of both precision, as well as F1-Score. Visual inspection of the detection

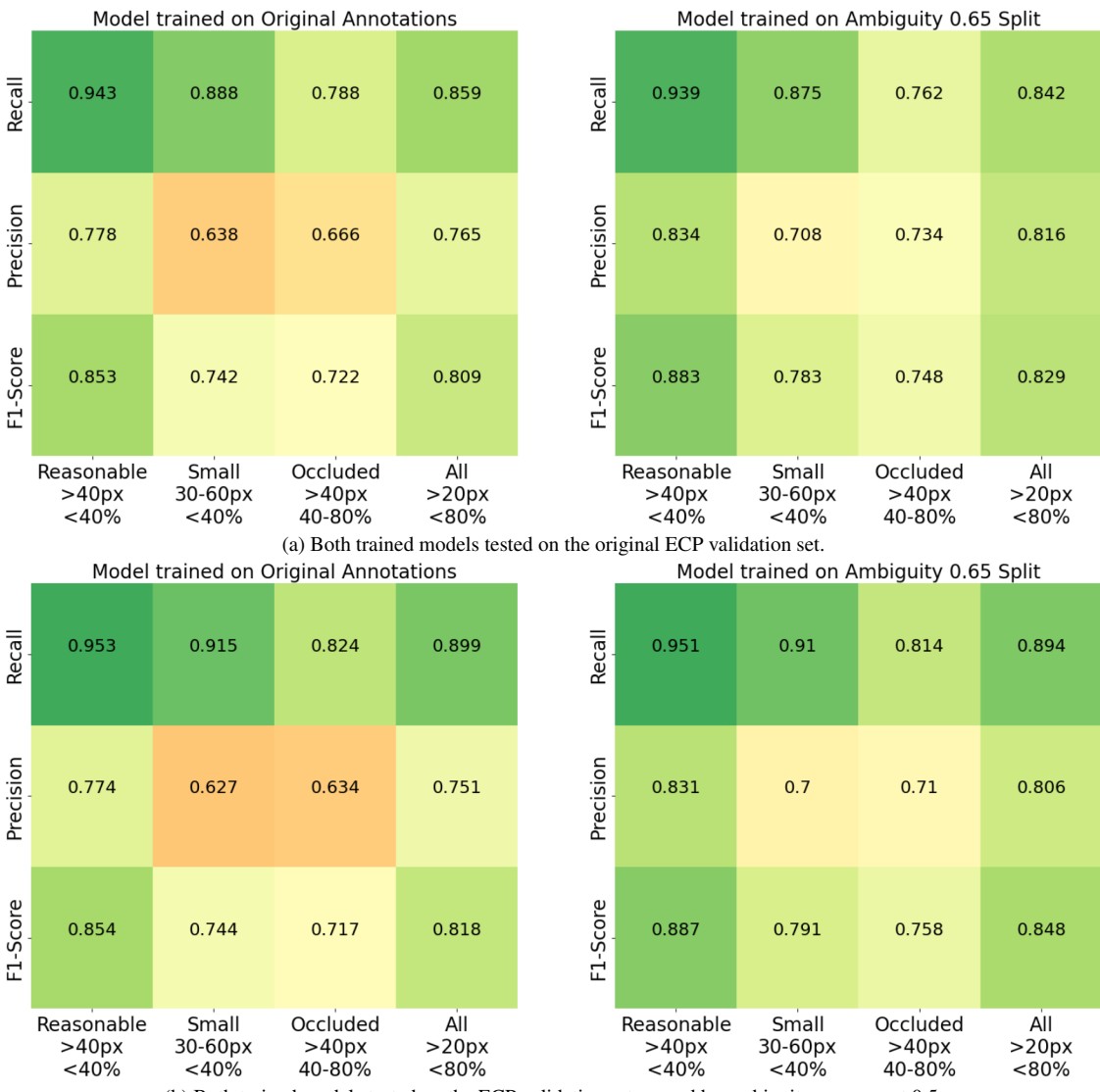

(a) Both trained models tested on the original ECP validation set.

(b) Both trained models tested on the ECP validation set pruned by ambiguity measure at 0.5.

Figure 4. Comparison of Recall, Precision and F-1 Score for two different training and test datasets.

errors confirmed that ambiguous data in the training set contributes to the generation of false positive detections. This trend is observable regardless whether the model was tested on data including or excluding ambiguous data. The recall slightly declines in all testing scenarios when the model is trained without the ambiguous instances, most notably for the "occluded" test subset. This might indicate that some of the removed ambiguous instances still convey information, which can help the model learn more diverse representations, especially in the presence of occlusion.

Note that, as can be expected, removing ambiguous data from the test set improves all metrics for both trained models. Nonetheless, the implication is less obvious: You can ignore ambiguous data in both sets, resulting in reduced cost

through lower training times. Simultaneously, annotation costs can be reduced, because it is possible to estimate the ambiguity measure reliably for high-ambiguity instances, and thereby exclude them from the annotation process all together [18].

**There is a strong correlation between ambiguity and occlusion.** When comparing the ambiguity measure with the occlusion tags in the ground truth (see Figure 5), we observe that higher values of the ambiguity measure correspond to a greater prevalence of occlusion tags within the dataset. As ambiguity increases, the proportion of tags indicating higher levels of occlusion is also elevated. This is evident in Figure 5, where the peak of the"occluded > 80"

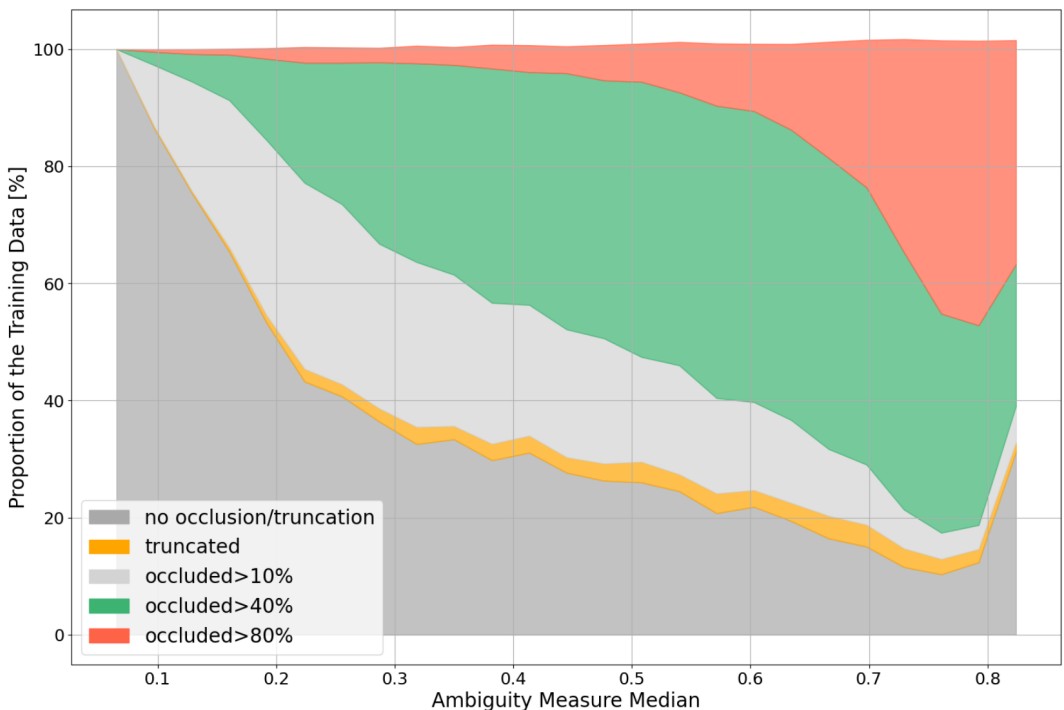

Figure 5. Distribution of occlusion and truncation tags for different ambiguity thresholds.

tag proportion is at an ambiguity measure value of 0.79. This explains why the model trained with applied ambiguity threshold, which achieves higher performance in all other evaluation subsets in terms of LAMR, is surpassed by the model trained on the original training set including all ambiguous data for the occluded subset (see Figure 3). When removing ambiguous instances, we disproportionately remove occluded instances. Hence, the model which has seen more occluded data in training performs better on this specific subset, while is still exhibits lower performance on all other data.

## 5. Improving Model Performance at Reduced Training and Annotation Costs

Based on the findings detailed above, we propose the following course of action for treating ambiguity in machine learning datasets, especially for safety-critical applications:
**1. Assess** possible sources of ambiguity in the labelling guide. Deriving simple rules, which are easy to communicate and cover the most important edge cases can help reduce ambiguity during the annotation process without adding possible sources of errors through excess difficulty for the annotators.
**2. Quantify** the ambiguity within the data. This can be done from the raw annotator answers or by estimation from the labelled data. Choose a method which is cost-efficient, as well as a quantitative measure which is appropriate for

your use-case and interpretable, *e.g.* by providing a ranking of the instances w.r.t. ambiguity.
**3. Inspect** a subset of the labelled results visually at different ambiguity thresholds. Examine the distributions of the ambiguity measure over different classes and intra-class properties to identify possible common sources of ambiguity. Determine if certain properties are over-represented at higher ambiguity thresholds. If annotators disagree over instances where the correct label seems obvious, this can possibly be amended by updating the labelling instructions.
**4. Prune** the dataset by removing highly ambiguous data up to a threshold determined through the previous steps. If the dataset is in danger of loosing representativeness, this can then be addressed through adapted data collection protocols or augmentation at training time.

## 6. Conclusion

As we have seen, we will always encounter some degree of ambiguity in annotated data. Additionally, the described experiments demonstrate that the prevalence of ambiguous data has implications for a machine learning model during both, training and testing. Our experiments show that we can improve the performance of a state-of-the-art detection model by simply removing ambiguous data to a certain extend. When doing so, we can identify two trade-offs, which need to be considered. Firstly, the very common trade-off in machine learning between recall and precision is also at

play when adding or removing ambiguous instance from training data. Secondly, when removing too many ambiguous instances the dataset is at risk of loosing representativeness. Therefore, an understanding of ambiguity in the dataset is important to decide which instances to remove, which to keep and which cases of hard-to-detect objects might be in the need of additional treatment to prevent them from being underrepresented in the remaining training set. As we have shown, a simple ambiguity measure, which can be estimated or calculated from the raw annotation answers of multiple workers, enables us to prune the dataset, resulting in improved model performance at reduced costs.

## 7. Future Work

Important topics for future work are the expansion of this framework onto different object classes as well as model architectures. We employed only one heuristic measure for ambiguity based on annotator answer frequencies for our evaluation. In future work, different measures to calculate and estimate ambiguity, including more elaborate techniques, should be investigated and compared w.r.t. how well they reflect ambiguity and are apt to provide a threshold for improving model performance by pruning the dataset.

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
