# OpenReview forum: "Ambiguous Annotations: When is a Pedestrian not a Pedestrian?"
_thecvf.com/CVPR/2024/Workshop/VLADR — VLADR 2024 Poster_

### Official Review · Reviewer_7mqL · 2024-04-21

**Rating:** 6
**Confidence:** 3

**Review:**

This paper studies the annotation quality issue of autonomous driving datasets, particularly focusing on pedestrians. The results indicate that excluding highly ambitious pedestrian annotations can both reduce annotation costs and improve models' detection performance. It is an interesting topic to include in our workshop. It would be even more interesting if the paper could extend this study to vision language models and see how "hallucination" can affect this issue.

---

### Decision · Program_Chairs · 2024-04-22

Accept (Poster)